# Effect of Fishmeal Content in the Diet on the Growth and Sexual Maturation of Olive Flounder (*Paralichthys*
*olivaceus*) at a Typical Fish Farm

**DOI:** 10.3390/ani11072055

**Published:** 2021-07-09

**Authors:** Su-Jin Park, Bong Seung Seo, Hung Sik Park, Bong-Joo Lee, Sang-Woo Hur, Taek-Jeong Nam, Kyeong-Jun Lee, Seunghyung Lee, Youn Hee Choi

**Affiliations:** 1Department of Fisheries Biology, Pukyong National University, Busan 48513, Korea; sujin_park@pukyong.ac.kr (S.-J.P.); smail5963@naver.com (B.S.S.); 2AlphaAqua, Busan 47042, Korea; hspak@chol.com; 3Aquafeed Research Center, National Institute of Fisheries Science, Pohang 37517, Korea; bonglee@korea.kr (B.-J.L.); maverickhuh@korea.kr (S.-W.H.); 4Institute of Fisheries Sciences, Pukyong National University, Busan 46041, Korea; namtj@pknu.ac.kr; 5Department of Marine Life Science, Jeju National University, Jeju 63234, Korea; kjlee@jejunu.ac.kr; 6Department of Marine Bio-Materials & Aquaculture, Pukyong National University, Busan 48513, Korea

**Keywords:** olive flounder, fishmeal replacement, typical fish farm, growth, maturity, hormones

## Abstract

**Simple Summary:**

Increasing demand for an efficient and economic fishmeal feed for sustainable aquaculture has urged the aquafeed sector to seek an optimum fish-feed formulation. This study investigated the physiological response in olive flounder fed various fishmeal diets in a typical fish farm. The fish were farmed for 20 weeks, using the following experimental feeds: a control feed (CON), a replacement by 20% (F20), and 30% (F30) of the fish meal content of the CON. All groups showed no significant difference in growth and survival rates. However, due to investigating hormone expression associated with maturation, high expression of PSS-I and low expression of FSH-β, ER-α, and ER-β in FM30 compared to other experimental groups were observed. Therefore, up to 30% fishmeal replacement does not affect growth, but it appears to have a slight effect on the sexual development of olive flounder.

**Abstract:**

Olive flounder (*Paralichthys olivaceus*) is a commercially important and valuable species for aquaculture in Korea. Due to the unstable supply of fishmeal for farmed fish, an optimum fish-feed formulation should be researched to ensure the sustainability of *P. olivaceus* aquaculture. This study investigated the effect of three experimental diets: Con (basal diet); FM_20_ (20% fishmeal replacement of CON); and FM_30_ (30% fishmeal replacement of CON) on *P. olivaceus* over 20 weeks at a typical farm by monitoring the growth and factors relating to sexual maturation. The results showed that no differences in growth were observed between the CON and diet-replacement groups. Gonadal oocyte development was similar between the CON and diet-replacement groups. Moreover, sbGnRH and GH expression did not differ between the CON and diet-replacement groups. The levels of Erβ and Vtg expression were significantly higher in the FM_20_ group than in the CON and FM_30_ groups after the experimental period. The expression of PSS-I was significantly higher in the FM_30_ group than in the CON and FM_20_ groups. Therefore, although growth occurred when 30% of the fishmeal was replaced, such high dietary protein replacement may be ill-advised during the maturation of olive flounder at the commercial fish farm.

## 1. Introduction

Olive flounder (*Paralichthys olivaceus*) is the most cultured fish in Korea, accounting for 43,320 tons, or 51% of the 85,217 tons of the total fish-farming production in 2019 [1]. It is considered an important and economically valuable aquaculture fish in Korea and Japan [2,3]. Thus, it is essential to secure healthy and mature broodstock during the production of juvenile fish to support production [4,5]. Feed is the major factor influencing the early growth, gonadal development, and sexual maturation of cultured olive flounder. High protein proportion in feed is essential, allowing the fish to obtain amino acids to build up new protein and use it to grow [6]. Therefore, fishmeal is considered the most optimum choice for the adequate protein supply and many other advantages are reported, e.g., a rich source of taurine, minerals (i.e., phosphorous), and vitamins without anti-nutritional factors [7]. Due to the continuous development of aquaculture in recent decades, a considerable demand for fishmeal is required for aquafeed. However, the supply of fishmeal is finite, making it necessary to reduce the levels of fishmeal in seeking an efficient and economic ratio of feed for farmed fish [8,9,10]. Although the growth and feed efficiency of olive flounder fed a proper fishmeal replacement are similar to those of fish fed traditional feed [11,12], research on the growth and sexual maturation associated with different food sources remains insufficient. Thus, recent studies on fish feed have attempted to substitute fishmeal and determine appropriate replacement rates.

Hormones synthesized and secreted by the endocrine system in fish play an important role in reproduction, growth, development, and metabolism [13]. Fish gonads develop under the influence of endocrine hormones produced by the hypothalamic-pituitary-gonadal axis [14]. The hypothalamus coordinates signals from environmental stimuli to produce kisspeptin (Kiss), a ligand of the G-protein coupled receptor GPR54, and Kiss stimulates gonadotropin-releasing hormone (GnRH) neurons to produce GnRH [15]. This hormone plays a major role in the reproduction of fish with 14 identified forms isolated in vertebrates [16]. Among them, it is proved that a species-specific variant, seabream-type GnRH (sbGnRH), takes responsibility for gonadotropin secretion and has a connection with gonadal maturation [3]. In the hypothalamus, preprosomatostatin (PSS), which is a somatostatin precursor protein, is produced. PSS suppresses the secretion of growth hormone (GH) and participates in sex steroid hormone activity and reproduction [17,18]. The pituitary gland produces the gonadotropins follicle-stimulating hormone (FSH) and luteinizing hormone (LH) under the influence of sbGnRH transmitted from the hypothalamus [16]. FSH and LH consist of α and β subunits. The α subunit confers a structural identity, whereas the characteristics of each hormone are determined by the β subunit [19]. FSH and LH synthesized in the pituitary are transported through the bloodstream to the gonads. GH is a growth-promoting hormone synthesized in the pituitary gland under the influence of sbGnRH secreted by the hypothalamus [16]. GH affects somatic cell development by acting on all cells and tissues [20]. Vitellogenin (Vtg) is produced in the liver of female fish upon activation by sex hormones and accumulates in the yolk [21]. The estrogen receptor (ER) is a 17β-estradiol (E2) receptor that induces Vtg synthesis in the liver. Vtg accumulates in oocytes and becomes an early nutritional source for developing embryos.

In this study, an experiment was conducted on olive flounder fed a diet with a low content of fishmeal instead of the high-fishmeal-content diet used on fish farms in Korea. This experiment was conducted at a typical commercial fish farm not at the laboratory level. The effects of a low-fishmeal-content diet on the sexual maturation of female flounder were investigated by monitoring the development of reproductive organs and identifying the levels of endocrine hormones in the hypothalamus, pituitary gland, and liver.

## 2. Materials and Methods

### 2.1. Experimental Fish, Feeding Conditions, and Sample Collection

A total of 28,200 *P. olivaceus* (initial mean weight 150 g) were randomly distributed into six 10 m × 10 m concrete tanks (depth: 60 cm and volume: 60,000 L) at a stocking density of 4700 fish/tank (*n* = 2 tanks per treatment) at a facility in Seongsan-eup, Seogwipo-si, Jeju-do, Republic of Korea. Each duplicate group of tanks were randomly assigned to one of three experimental diets. The fish were hand-fed twice a day (08:00 and 18:00 h) at the feeding rate (% body weight per day) ranging from 0.42 to 1.46 for 20 weeks. A flow-through culture system using the mixture of seawater (35 ppt) and underground seawater (29 ppt) at the ratio of 7:3, respectively, was used. The rearing water exchange rate was 2000% per day (i.e., whole water in the tank was exchanged completely 20 times per day) (see Figure 1). The water quality of the tanks was maintained as follows: salinity, 31 ± 1.5 (mean ± SD) ppt; water temperature, 18.5 ± 1.01 °C; pH, 8.02 ± 0.04; and dissolved oxygen, 7.08 ± 0.06 ppm. The uneaten feed was removed at 1 h after every meal to calculate feed efficiency, and survival was monitored daily during the feeding experiment.

To assess growth and maturation, 20 fish (24 h post starvation) were randomly selected from each experimental group at 0, 8, 16, and 20 weeks and euthanized with 100 ppm of 2-phenoxyethanol. Total length (cm), total weight (g), body height (cm), body width (cm), and gonad weight (g) were measured. Tissues samples were dissected from these fish and fixed for histological (gonads) and qPCR (brain, pituitary, liver) analyses, accordingly. The equations used for calculation of growth performance, including weight gain (%), specific growth rate (%/day), condition factor, feed conversion ratio, protein efficiency ratio, feed intake (g), and survival (%) is provided as follows:Weight gain (%) = (FBW − IBW)/IBW × 100
Specific growth rate (%/day) = (Ln(FBW) − Ln(IBW))/days of feeding × 100
Condition factor = FBW/TL^3^ × 100
Feed conversion ratio = Dry feed fed/(FBW − IBW)
Protein efficiency ratio = (FBW − IBW)/protein given
Feed intake = Total dry feed consumed/number of fish
Survival (%) = Final fish number/initial fish number × 100
where IBW, FBW, and TL were initial mean body weight (g), final mean body weight (g), and total length (cm), respectively. To analyze changes in whole-body proximate composition of the flounders fed the test diets, after 20 weeks of experiment, additional 20 fish per tank were randomly selected after 20 weeks, euthanized with 100 ppm of 2-phenoxyethanol, and kept at −20 °C until analysis. The samples were pooled and grounded, after the freezing-fry and were subjected to the proximate composition two tanks per treatment). The analysis was performed following, an AOAC [22] method. In brief, moisture was analyzed using the heating and drying method (125 °C, 3 h), and ash was analyzed using the classic dry ashing technique (550 °C, 4 h). Proteins were determined via automated crude protein analysis (Kjeltec 2300, FOSS, Hillerød, Sweden). Lipids were analyzed according to the method in Folch et al. [23].

### 2.2. Experimental Diet Preparation

In the control diet (CON) used in the experiment, no fishmeal was replaced. The composition of the basal diet is shown in Table 1. Sardine fish meal, anchovy fish meal, tankage meal, poultry by-product meal, soybean meal, wheat gluten, and soy protein concentrate were used as the feed protein sources (Table 2). Flour was used as the carbohydrate source, and fish oil was the lipid source. The contents of crude protein and crude lipid were designed to be the same in all experimental feeds. Three experimental diets were tested: a basal diet in which none of the fishmeal was replaced, which was the CON diet, a diet in which 20% of the fishmeal in the CON diet was replaced (FM_20_), and a diet in which 30% of the fishmeal in the CON diet was replaced (FM_30_). Analysis of the general components of the fish feed was performed using the same method described earlier.

### 2.3. Male and Female Identification and Histology of Oocyte Development

To observe gonadal development, we used methods as previously described by Choi et al. [24]. Gonads were fixed in Bouin’s solution (Sigma–Aldrich, Mannheim, Germany) for 24 h, and then in distilled water, and stored in 70% ethanol. The fixed tissues were embedded in paraffin to prepare tissue blocks for histology. The glass slides were dyed in a deparaffinization step using xylene. After the gonadal tissues were stained with hematoxylin and eosin, male and female gonads were distinguished. The results of the sex ratio of 20 random fish from each experimental group were displayed in Table 6. To assess the oocyte development, ten random ovarian tissues from each group were selected and observed using an optical microscope (Olympus BX41, Tokyo, Japan). Furthermore, to calculate the diameters (μm) of the oocytes, each oocyte from 100 random ones of ten selected female was measured ten times by Motic Image Plus 2.0 (Motic Instruments Inc., Xiamen, China) and the mean value represented for the diameters of the oocyte, following Bateman et al. [25] and Ju and Lee [26].

### 2.4. Expression of Endocrine Hormones Related to Sexual Maturation

According to the result of the sex ratio from three experimental groups (Table 6), The minimum number of females was ten belong to the F30 group at week 16. Hence, ten female *P. olivaceus* will be applied to all hormone-related experiments in this investigation.

#### 2.4.1. Total RNA Extraction

Ten female *P. olivaceus* of each experimental group from week 0 and week 20 were selected for this investigation. Their brain, pituitary, and liver tissues were extracted and homogenized by adding 600 µL of TransZol UP (TransGen, Beijing, China), and the mixtures were reacted at 4 °C for 2 h. After the reaction was completed, 200 µL of chloroform was added to each sample, and the sample was mixed for 30 s and left to react at room temperature for 10 min. The tubes were centrifuged at 4 °C for 10 min to isolate the supernatant, after which 200 µL of supernatant was added to a 1.5-mL tube, 200 µL of isopropanol was added, and the solution was mixed for 30 s and left to react for 10 min at room temperature. The RNA pellet was separated by centrifugation at 4 °C for 10 min. The RNA pellet was washed with 600 µL of 75% ethanol diluted in DEPC water, then with 600 µL of 100% ethanol diluted in DEPC water, and dried completely in a 37 °C oven for 15 min. Thereafter, the RNA pellet was dissolved in 10–50 µL of RNase-free dH_2_O and stored at −75 °C. Total RNA was quantitatively and qualitatively analyzed by measuring RNA at A260/280 nm in the range of 1.8–2.0 using the NEO-Nabi UV/Vis NANO Spectrophotometer (NEOGEN, Lansing, MI, USA).

#### 2.4.2. cDNA Synthesis

Total RNA was diluted to 1 µg as a template, and cDNA was synthesized using a PrimeScript^TM^ 1st Strand cDNA Synthesis kit (Takara, Kyoto, Japan). The dNTP mixture and the Oligo dT primer were added to the diluted total RNA, and the mixture was left to react at 65 °C for 5 min. The temperature was lowered to 4 °C, and 5× PrimeScript Buffer, RNase inhibitor, and PrimeScript RTase were added to the reaction mixture, which was left to react at 42 °C for 60 min, and RTase activity was stopped at 95 °C for 5 min. DNase-free dH_2_O (180 µL) was added to dilute the mixture to 100 ng µL^−1^.

#### 2.4.3. Oligonucleotide Primer Production

Forward and reverse primers for 18S ribosomal, Kiss-I, sbGnRH, PSS-I, GH, FSH-β, LH-β, Vtg-I, ERα, and ERβ were used in the experiment to amplify 100–250 bp of the base sequence (Table 3). Fabrication was performed by Macrogen (Seoul, Korea), and the prepared primers were diluted to 10 pmoles in Tris-EDTA buffer and used for reverse transcription-polymerase chain reaction (RT-PCR).

#### 2.4.4. Real-Time qPCR (qPCR) Analysis

EmeraldAmp^®^ GT PCR Master Mix (Takara, Koyto, Japan) was used for qPCR with the following program: initial denaturation (94 °C, 5 min), one cycle; denaturation (94 °C, 30 s), annealing (55 °C, 30 s), and extension (72 °C, 30 s), 30 cycles; and final elongation (72 °C, 7 min), one cycle. The amplified qPCR product was electrophoresed for 25 min at 100 V in a 1.5% agarose/Tris-acetate-EDTA gel containing 7.5% ethidium bromide. Bands were confirmed using an Azure Biosystems instrument (C300, Azure Biosystems Inc., Dublin, CA, USA). Thereafter, 18s rRNA was used as a reference gene in relative quantification analysis with Gene Tools version 4.03 (SYNGENE, Cambridge, UK).

### 2.5. Statistical Analysis

All results are expressed as the mean ± standard error. One-way analysis of variance was performed using IBM SPSS Statistics version 25 software (IBM SPSS, Armonk, NY, USA). Duncan’s multiple range test was used to detect differences between means. A *p*-value < 0.05 was considered to indicate significance.

## 3. Results

### 3.1. Growth Performance and Gonadosomatic Index (GSI) Analysis

The growth performance results are shown in Table 4. No significant differences in weight gain, specific growth rate, condition factor, feed intake, and survival (*p* > 0.05) were observed between the CON and experimental groups. The feed conversion ratio did not differ between the CON and experimental groups (*p* > 0.05), and the feed conversion ratio of the FM_30_ group was significantly lower than that of the FM_20_ group (*p* < 0.05). The protein efficiency ratio did not differ between the CON and experimental groups (*p* > 0.05), but the protein efficiency ratio of the FM_30_ group was significantly higher than that of the FM_20_ group (*p* < 0.05).

Table 5 shows the whole-body analysis results. From the results, lipid contents in the FM_20_ and FM_30_ groups were significantly higher than that in the CON group (*p* < 0.05), and the FM_30_ group had significantly lower ash content than that of the CON group. No significant differences in moisture or protein content were observed between the CON and experimental groups (*p* < 0.05).

The GSI from all experimental groups tended to increase with body weight, and the CON group had a significantly higher value after 16 weeks (*p* < 0.05). The FM_20_ group had a significantly higher GSI after 8 weeks (*p* < 0.05), and the FM_30_ group had a significantly higher GSI after 16 weeks (Figure 2, *p* < 0.05).

### 3.2. Male and Female Identification and the Histology of Oocyte Development

The sex ratio of 20 randomly selected flounders from each group was determined during each experimental week. The sex ratios in the CON, FM_20_, and FM_30_ groups at 8, 16, and 20 weeks are shown in Table 6. There were more females than males. From assessments of gonadal development, a large number of nucleoli was observed in the nucleus as the size of the oocyte nucleus increased in all experimental cells (Figure 3). The 20-week oocytes in the FM_20_ group were significantly larger than those in the CON and FM_30_ groups (*p* < 0.05), but overall, the size of the oocytes increased over time, with no significant differences at the other experimental timepoints (Figure 4, *p* > 0.05).

### 3.3. Expression of Endocrine Hormones Related to Sexual Maturation

According to the sex ratios in the CON, FM_20_, and FM_30_ groups at 8, 16, and 20 weeks, the minimum number of female olive flounder is ten distributed in FM_30_ groups. To have an equal comparison, the number of female olive flounder used in each experiment in this investigation was ten.

#### 3.3.1. Hypothalamic Kiss-I, sbGnRH, and PSS-I mRNA Expression

The expression of hypothalamic hormones related to sexual maturation was determined at the beginning and end of the experiment. The expression of Kiss-I, which affects sbGnRH synthesis by modulating external environmental stimuli in the hypothalamus, did not differ significantly in all experimental groups (Figure 5, *p* > 0.05). sbGnRH expression in the CON group did not differ significantly from the initial level and that in the experimental groups (*p* > 0.05). The expression of PSS-I, which affects growth and maturation, was significantly higher in the FM_30_ group compared to initial levels and the CON and FM_20_ groups (*p* < 0.05). No significant difference was observed between the CON and FM_20_ groups (*p* > 0.05). The initial PSS-I expression level was significantly lower for the CON group and both experimental groups (*p* < 0.05).

#### 3.3.2. Pituitary GH, FSH-β, and LH-β mRNA Expression

Pituitary GH expression did not differ significantly from the initial level and between the CON and experimental groups (Figure 6, *p* > 0.05). FSH-β expression was significantly higher in the FM_20_ group compared to the initial level and the CON and FM_30_ groups (*p* < 0.05), and levels in the CON and FM_20_ groups were higher compared to the initial level (*p* < 0.05). The expression of LH-β was did not differ significantly in all experimental groups compared to the initial level (*p* > 0.05).

#### 3.3.3. Liver ERα, ERβ, and Vtg mRNA Expression

Hepatic expression of ERα was significantly higher in the FM_20_ group compared to the initial level and the FM_30_ groups (Figure 7, *p* < 0.05). ERβ expression was significantly higher in the FM_20_ group compared to the initial level and the CON and FM_30_ groups (*p* < 0.05), and the lowest values were observed for the initial group (*p* < 0.05). The initial Vtg level was significantly lower than those in the CON group and both experimental groups (*p* < 0.05). No significant difference in Vtg level was detected between the CON and FM_30_ groups (*p* > 0.05), and the FM_20_ group had a significantly higher level compared to the initial level and the CON and FM_30_ groups (*p* < 0.05).

## 4. Discussion

Fishmeal usually comprises approximately 60% or more of aquaculture feed, and research to replace fishmeal has been conducted over a long time [27]. Lim et al. [28] reported that up to 50% of the fishmeal can be replaced when the replacement is properly blended with concentrated soy protein, wheat gluten, and poultry by-products and added to feed. Fish growth and biomarkers are monitored to analyze the growth rate of target species in feeding experiments. In this study, the experimental fish fed less fishmeal grew at the same rate as the control fish and exhibited similar or good feed conversion and protein conversion efficiency. This was similar to the results of studies on the growth of *P. olivaceus* [12] and the starry flounder *Platichthys stellatus* [29] in individuals fed a diet low in fishmeal content. In this study, no difference in whole-body protein content was observed, so we believe that reducing an appropriate amount of fishmeal in the feed did not affect protein composition. Liver and muscle protein increase in sea bass (*Sebastes schlegelii*) [30] and mudskipper (*Boleophthalmus pectinirostris*) during ovarian maturation [31]. In this study, the gonads and GSI increased during the growing season as well. Ovarian development in teleost fish is subdivided into synchronous, group-synchronous, and asynchronous types [32]. Flounder undergo an asynchronous type of development in which early populations of oocytes exist simultaneously with mature oocytes. In this study, the size of the nucleus of oocytes increased sequentially and numerous basophilic nucleoli at the edge of nuclei surrounding by uniformed-stain cytoplasm were observed in all experimental groups at 8 and 16 weeks. At the period of 20 weeks, in accompanying with oocyte growth, the nuclei and nucleoli increase in size with less densely stained cytoplasm and invisibility of zone radiata at all experimental groups. Ovarian development can be divided into growing, maturing, ripening and spawning, recovery, and rest phases. In the CON and experimental groups, all flounder ovaries had characteristics of growth to maturation in this study, suggesting that reducing the fishmeal content by 30% did not affect gonadal development. Sexual maturation in Osteichthyes occurs under endocrine control [33].

Kiss-I expression in the hypothalamus was found to be higher during the mature state than during the immature state in Hongbari (*Epinephelus fasciatus*) [34], as well as in mammals such as the Siberian hamster (*Phodopus sungorus*) [35]. In one maturation induction experiment, the weight of the gonads increased, and the expression of Kiss-I also increased once maturity was reached. Kiss-I is associated with food intake in addition to reproduction. In female rats (*Rattus*), expression of the Kiss-I gene [36] and the Kiss receptor gene decreased during and at 18 h after fasting [37]. In this study, the expression of Kiss-I and GSI conflicted with the outcomes expected from food competition among individuals during fasting, sample collection, and mass rearing in a farm environment. sbGnRH induced FSH and LH production by activating the sbGnRH receptor located in the anterior pituitary gland [38] of the black Porgy, *Acanthopagrus schlegeli* [39] and the gilthead seabream, *Sparus aurata* [40]. In this study, sbGnRH mRNA expression did not differ significantly between the CON and experimental groups (*p* > 0.05) but exhibited a similar trend with Kiss-I mRNA expression. FSH stimulates E2 production by oocytes and is involved in early gamete formation and gonadal growth. LH induces ovulation [41,42,43]. The expression level of FSH-β was significantly higher in the FM_20_ and CON group (*p* < 0.05). sbGnRH secretion promotes the secretion of FSH-β and LH-β, and although there was no significant difference in sbGnRH level in this study, it was confirmed that FSHβ, which is involved in early maturation, was expressed significantly higher in FM20. There was no significant difference in the expression of LH-β in all groups because it was only in the maturation phase and not yet in the ovulation phase [44]. The ER subtypes are regulated by ERα and ERβ. In mammals, ERα is mainly distributed in the mammary gland, ovaries, bone, and liver tissue, whereas ERβ is distributed in the prostate, ovaries, and adipose tissue, and both affect ovarian development [45]. Vtg is synthesized upon E2 binding to the ER [46]. Vtg is produced in the liver under the influence of E2 and the ER. Vtg is a high-molecular-weight protein with attached lipids, sugars, and phosphates, which are taken up by oocytes and used as nutrition for embryonic development after fertilization [47]. The results of our experiment showed that the expression of ERα was high, along with high Vtg levels in the FM_20_ group. PSS-I, expressed in the hypothalamus, is a somatostatin precursor protein that inhibits the expression of GH in the pituitary gland [18] and is influenced by hormones during puberty. It was reported that PSS-I was inhibited in the hypothalamus when sex hormones were injected into rainbow trout (*Oncorhynchus mykiss*) [48]. GH secretion increases during puberty [49]. GH increased in the blood with gonadal development in *O. niloticus* [50]. However, sex hormones did not affect GH RNA expression in *C. auratus* or the fathead minnow *Pimephales promelas* [51,52]. This is an example of species specificity. In the present study, PSS-I increased as the substitution rate of fishmeal increased, and trends in its expression contrasted with those of FSH and LH expression. GH expression did not differ between the CON and experimental groups, but it exhibited a downward trend, as with PSS expression. Thus, PSS-I affected GH, suggesting that PSS-I and GH expression are affected by the progression of puberty. Taken together, sex steroid hormone production and reproductive activity were significantly higher in the FM_20_ group than in the CON and FM_30_ groups.

Therefore, although growth was not affected by a fishmeal substitution rate of 30%, diet replacement may be ill-advised during the early puberty stage of *P. olivaceus*.

## 5. Conclusions

Up to 30% fishmeal replacement in the composition of the basal diet in this study does not affect growth, but it has been shown that there are some low-grade expressions of mature related hormones. Therefore, to use low-fishmeal diet in the field, it is believed that the most appropriate proportion of fish content should be selected by examining the metabolic and physiological responses in olive flounder as well as the growth of fish due to reduced fish meal content.

## Figures and Tables

**Figure 1 animals-11-02055-f001:**
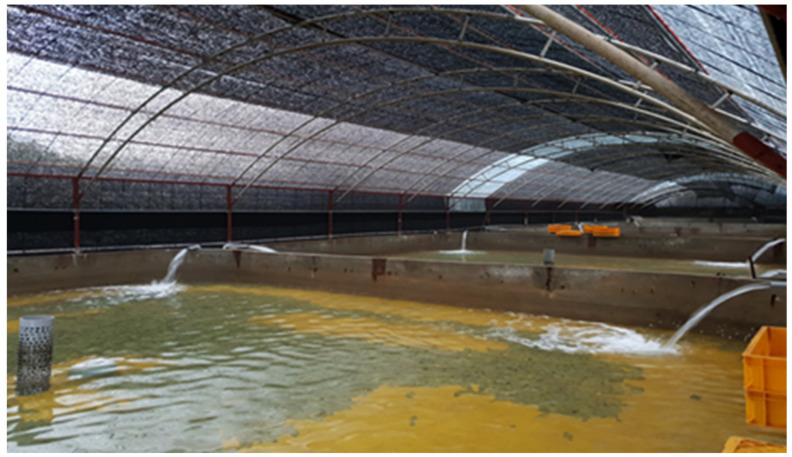
Picture of the flow-through tank systems at a land-based facility used for the current study.

**Figure 2 animals-11-02055-f002:**
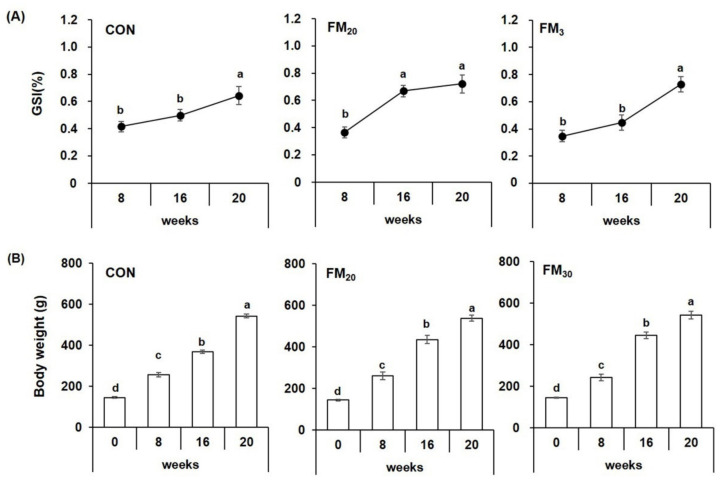
Gonadosomatic index (GS (**A**)) and total body weight (**B**) of the olive flounder *Paralichthys olivaceus.* Data are presented as the mean ± SEM (*n* = 20). Different letters above the bars indicate significant differences (*p* < 0.05) between groups at each sampling time point. CON is the control basal diet in which no fishmeal was replaced, FM_20_ is the diet with 20% of the fishmeal replaced, and FM_30_ is the diet with 30% of the fishmeal replaced.

**Figure 3 animals-11-02055-f003:**
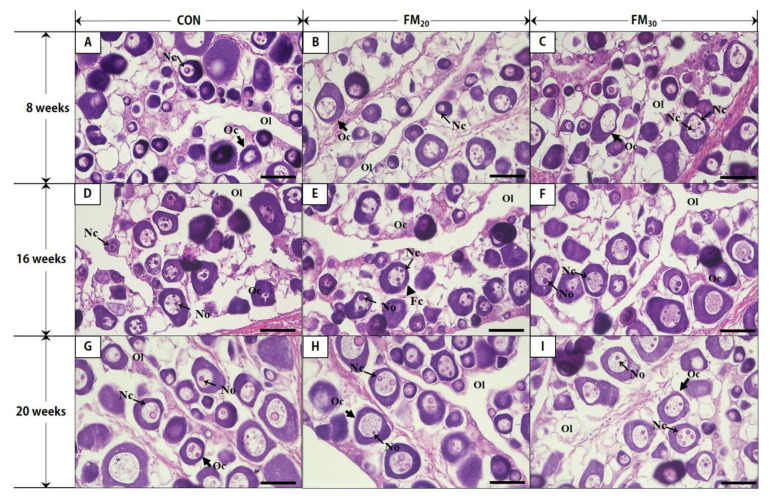
Hematoxylin and eosin-stained gonads of female olive flounders (*Paralichthys olivaceus*) fed low-fishmeal diets. (**A**–**C**), oocytes of 8 weeks (*n* = A − 18, B − 16, C − 15); (**D**–**F**), oocytes of 16 weeks (*n* = D − 15, D − 17, E − 10); (**G**–**I**), oocytes of 20 weeks (*n* = G − 14, H − 18, I − 17). CON, control diet; FM_20_, diet with 20% of the fishmeal replaced; FM_30_, diet with 30% of the fishmeal replaced. Oc, oocyte; Nc, nucleus; No, nucleolus; Fc, follicular cell; Ol, ovarian lumen. Scale bars = 50 μm.

**Figure 4 animals-11-02055-f004:**
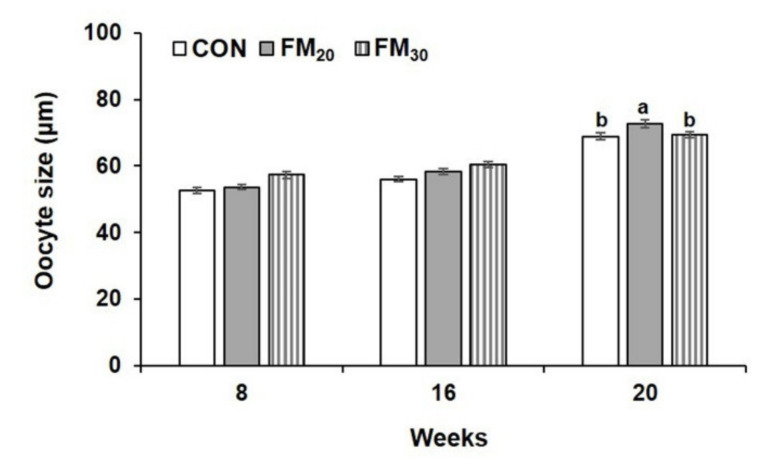
Oocyte diameters measured at 100× magnification at 8 weeks, 16 weeks, and 20 weeks. Data are presented as the mean ± SEM (*n* = 10). Different letters above the bars indicate significant differences (*p* < 0.05) between groups. CON, control diet; FM_20_, diet with 20% of the fishmeal replaced; FM_30_, diet with 30% of the fishmeal replaced.

**Figure 5 animals-11-02055-f005:**
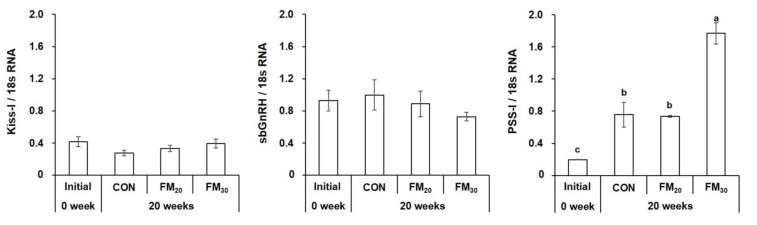
Expression of kisspeptin (Kiss) I, gonadotropin-releasing hormone (sbGnRH), and preprosomatostatin (PSS) I mRNA in the olive flounder *Paralichthys olivaceus.* Data are presented as the mean ± SEM (*n* = 10). Different letters above the bars indicate significant differences (*p* < 0.05) between groups. CON, control diet; FM_20_, diet with 20% of the fishmeal replaced; FM_30_, diet with 30% of the fishmeal replaced.

**Figure 6 animals-11-02055-f006:**
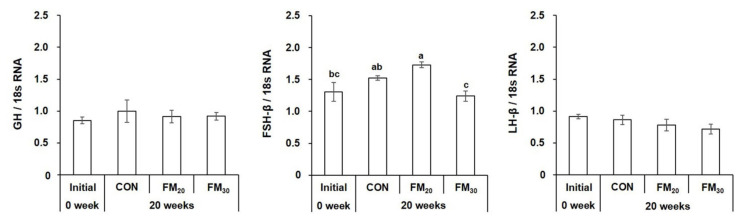
Expression of growth hormone (GH), follicle-stimulating hormone (FSH)-β, and luteinizing hormone (LH)-β mRNA in the olive flounder *Paralichthys olivaceus.* Data are presented as the mean ± SEM (*n* = 10). Different letters above the bars indicate significant differences (*p* < 0.05) between groups. CON, control diet; FM_20_, diet with 20% of the fishmeal replaced; FM_30_, diet with 30% of the fishmeal replaced.

**Figure 7 animals-11-02055-f007:**
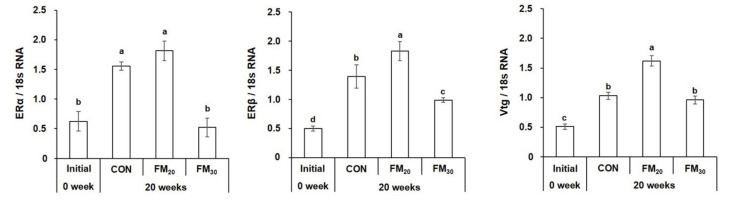
Expression of estrogen receptor (ER) α, ERβ, and vitellogenin (Vtg) mRNA in the olive flounder *Paralichthys olivaceus.* Data are presented as the mean ± SEM (*n* = 10). Different letters above the bars indicate significant differences (*p* < 0.05) between groups. CON, control diet; FM_20_, diet with 20% of the fishmeal replaced; FM_30_, diet with 30% of the fishmeal replaced.

**Table 1 animals-11-02055-t001:** Composition of the basal diet for the olive flounder *Paralichthys olivaceus* (%, dry matter basis).

Component	Diets		
	CON	FM_20_	FM_30_
Sardine FM	31.50	25.00	21.75
Anchovy FM	31.50	25.00	21.75
Soybean meal	12.00	12.00	12.00
Wheat flour	16.90	14.90	14.95
Wheat gluten	1.00	3.50	4.50
SPC	-	3.50	5.30
Tankage meal	-	3.50	6.25
Poultry by-product meal	-	3.50	4.00
Fish oil	3.40	4.00	4.30
Lecithin	0.50	0.50	0.50
Betaine	-	1.00	1.00
Taurine	-	0.40	0.50
Mono calcium phosphate	0.50	0.50	0.50
Mineral mix	1.00	1.00	1.00
Vitamin mix	0.80	0.80	0.80
Vitamin C	0.30	0.30	0.30
Vitamin E	0.30	0.30	0.30
Choline	0.30	0.30	0.30
Total	100	100	100
Proximate composition (%, dry matter basis)			
Crude protein	56.80 ^ns^	57.20	56.80
Crude lipid	8.96 ^ns^	9.29	9.66
Crude ash	11.3 ^ns^	10.80	10.80
Proximate composition (%, wet basis)			
Moisture	7.70 ^ns^	7.75	7.42
Crude protein	52.40 ^ns^	52.90	52.60
Crude lipid	8.27 ^ns^	8.59	8.94
Crude ash	10.40 ^ns^	10.00	9.97

FM, fishmeal; SPC, soy protein concentrate; ^ns^, not significantly different among groups.

**Table 2 animals-11-02055-t002:** International names of the protein sources used in this study.

Ingredients	International Name	International Feed Number
Fish meal, sardine	Fish, sardine meal, mechanical, extruded	5-02-015
Fish meal, anchovy	Fish, anchovy meal, mechanical, extruded	5-01-985
Soybean meal	Soybean meal, solvent, extruded	5-04-604
Wheat gluten	Wheat gluten meal	5-05-220
Soy protein concentrate	Soybean protein concentrate	5-08-008
Tankage meal	Meat and bone meal	5-00-388
Poultry by-product meal	Poultry by-product, meal rendered	5-03-798

**Table 3 animals-11-02055-t003:** Primers used for real-time qPCR (qPCR) amplification.

Primers	Sequence	Size	GenBank Accession
18s rRNA	F: 5′-GGTCTGTGATGCCCTTAGATGTC-3′	107 bp	EF126037.1
	R: 5′-AGTGGGGTTCAGCGGGTTAC-3′		
Kiss-I	F: 5′-AGCCACTTGTATCACCCTGA-3′	240 bp	KP347690.1
	R: 5′-GCCCTCTCGTGTGTTTTAGA-3′		
sbGnRH	F: 5′-AAATGGCTGTGAAGACCTTG-3′	150 bp	DQ074693.1
	R: 5′-CCTCAACTACATTGCCCAGA-3′		
PSS-I	F: 5′-ATGAAGATGGTGTCCTCCTCG-3′	174 bp	AB693833.1
	R: 5′-CTCGTTCCAGATCCACATGG-3′		
GH	F: 5′-GGAGGATCAACGTCTTCTCAA-3′	183 bp	D29737.1
	R: 5′-ACTGCGGCTGTTACTTATTCA-3′		
FSH-β	F: 5′-AGCTTCGACTGTCGTCCAAC-3′	140 bp	AB042422.1
	R: 5′-TGTTTAGCCGGACCTGTTTC-3′		
LH-β	F: 5′-CCGACGTGTCTTCTCATCAA-3′	136 bp	AB042423.1
	R: 5′-TGTTGAGGAAGGGGATCTTG-3′		
Vtg-I	F: 5′-TGAGCTCTTTGAGTACAGCG-3′	181 bp	AB200267.1
	R: 5′-TCCTCTCTGGATGTTCAGCA-3′		
ERα	F: 5′-ATGCTGGAGACTATCACTGACG-3′	113 bp	AB070629.1
	R: 5′-TGTCTGATGTGGGAGAGCAG-3′		
ERβ	F: 5′-ACCATCCAGGGAAACTCATC-3′	189 bp	AB070630.1
	R: 5′-GGCACATGTTGGAGTTTAGG-3′		

**Table 4 animals-11-02055-t004:** Growth performance, feed use, and morphological indices of olive flounders (*Paralichthys olivaceus*) fed experimental diets for 20 weeks.

Growth Performance	Experimental Groups		
	CON	FM_20_	FM_30_
Weight gain (%)	276.45 ± 11.5 ^ns^	273.94 ± 12.4	276.08 ± 12.4
Specific growth rate (%/day)	0.94 ± 0.02 ^ns^	0.93 ± 0.02	0.94 ± 0.02
Condition factor	1.11 ± 0.02 ^ns^	1.11 ± 0.02	1.12 ± 0.01
Feed conversion ratio	1.03 ± 0.04 ^ab^	1.11 ± 0.05 ^b^	0.95 ± 0.04 ^a^
Protein efficiency ratio	1.78 ± 0.07 ^ab^	1.64 ± 0.07 ^b^	1.92 ± 0.08 ^a^
Feed intake (g)	398 ± 54.7 ^ns^	422 ± 14.6	365 ± 10.1
Survival (%)	62.3 ± 2.32 ^ns^	62.5 ± 4.21	65.9 ± 1.37

Data are presented as the mean ± SEM for each group (*n* = 20). Values in the same row with different superscript letters are significantly different (*p* < 0.05). ^ns^, not significant.

**Table 5 animals-11-02055-t005:** Whole-body proximate composition of growing olive flounders (*Paralichthys olivaceus*) fed experimental diets for 20 weeks (% on wet basis).

Measurement	Diets		
	CON	FM_20_	FM_30_
Moisture (%)	70.0 ± 0.01 ^ns^	68.3 ± 1.19	69.4 ± 0.52
Protein (%)	21.3 ± 0.73 ^ns^	21.5 ± 0.58	21.9 ± 0.70
Lipid (%)	4.69 ± 0.19 ^a^	5.44 ± 0.15 ^c^	5.01 ± 0.14 ^b^
Ash (%)	3.40 ± 0.01 ^b^	3.14 ± 0.17 ^ab^	3.05 ± 0.09 ^a^

Values are means of duplicate groups of fish. Values in each row with different superscript letters are significantly different (*p* < 0.05). ^ns^, not significant.

**Table 6 animals-11-02055-t006:** Results of olive flounders (*Paralichthys olivaceus*) sex ratio from three experimental groups. F, female; M, male.

Weeks	Experimental Groups		
	CON (F:M)	FM_20_ (F:M)	FM_30_ (F:M)
8	9:1	8:2	7.5:2.5
16	7.5:2.5	8.5:1.5	5:5
20	7:3	9:1	8.5:1.5

## Data Availability

The data presented in this study are available on request.

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
