# Peer review of "Effect of Fishmeal Content in the Diet on the Growth and Sexual Maturation of Olive Flounder (Paralichthys olivaceus) at a Typical Fish Farm"

_animals, 2021, doi:10.3390/ani11072055_

Round 1

Reviewer 1 Report

This is an interesting study on the effects of fishmeal replacement in the diets for fish in aquaculture. Although not conceptually new, this study, performed under fish farming conditions, provides valid basic and applied information on potential effects of diet composition on growth and gonad maturation in fish and particularly in Japanese flounder, a relevant aquaculture fish species.

The results show no deleterious effects of 20 and 30% fishmeal replacement diets on growth (weight, SGR, condition factor, etc), survival and gonad development (oocyte size, GSI). On another hand, it shows through qPCR some effects of the experimental diets on the expression levels of a bunch of genes related to growth and reproduction, particularly, up-regulation of PSS-I and down-regulation of FSHbeta, ERalfa and ERbeta in the FM30 group, providing bases for potential negative effects of diets with more that 30% fishmeal replacement on fish welfare.

Although results are “potentially” interesting, as mentioned, they are weakened by important inaccuracies and concerns related to both methodologies and data, which should be corrected, as described in the following lines.

Major comments

  1. An important parameter analyzed in the study is the GnRH system. The authors mention the “GnRH” as the hormone under study, but the brain of most fishes, including the Japanese flounder (Fang et al. 2006, Pham et al. 2006,2007), express 3 GnRH genes giving rise to 3 GnRH peptides with different physiological roles. The authors must introduce and describe correctly this aspect and explain which gene they have analyzed.
  2. The methodology, data and presentation and discussion of results concerning “gonad maturation” are inaccurate and confusing and should be revised. In the discussion there are many lines on results obtained on the size of nucleus or content of phosphorus of the oocytes in the experimental groups, that are not shown in results, neither described in methodologies. Similarly, authors cite effects on gonad maturation, specifically the presence of mature oocytes, which we cannot see in the figures, neither in the text of the results section or a methodological description of how the authors analyzed oocyte stages of development. From the results of the study, we can see immature gonads in all groups through the experimental period, including the slight increase in GSI so, difficult to understand the discussion lines referring to potential effects on gonad maturation; no mature oocytes are evident in any group from the results presented.
  3. Another important concern is the analyzed sampling number (the “n”) for each analysis. It must be clearly indicated the “n” corresponding to the sampling points of each parameter. It is mentioned in methods that 20 fish were sampled at each time and they provide a wide estimation of the sex ratio. Then, they mention for several parameters that the “n” is 20, assuming that this analysis was performed in both males and females, which is correct. But, mainly in relation to the analysis of gene expression by qPCR, there is an important lack of information. Were all genes analyzed in both males and females and thus n=20?; probably not, as mentioned in a figure legend (citing n=6). If some analysis were performed only in one sex, this should be mentioned and the “n” included. It should be kept in mind that a low “n” might reduce the solidity of data, which would be rare considering that we work with experimental groups of thousands of fish; we can also question why hormone analysis were not performed at 8 and 16 weeks, which would provide much more information to the study. In any case, the authors should address this question and include all information in the text and figures.
  4. The presentation of the statistics of data, it is sometimes confusing or inaccurate, in figures and legends. For example, legend of fig 1 say that “different letters above the bars indicate significant differences between groups”, but groups are represented in different graphs and for each sampling point all data between groups are similar, making this representation unnecessary; and no mention to the meaning of the “*”, here and in other figures. With these inaccuracies it makes difficult to understand the data and the potential differences between sampling points or groups.
  5. In relation to the previous comment, there are inaccuracies in the text concerning some results, that is relevant to the whole information provided by the study. For example, in the abstract and other parts of the text, the authors say that FSHbeta and ERalfa are higher in FM20 than controls or that VTG expression in FM30 is lower than controls, which is not true. The authors must revise the text throughout, correct inaccuracies and re-write the text accordingly.

Minor comments

  1. The discussion should be improved, to include a more extensive discussion on relevant results that are very slightly considered, such as the different levels of FSHbeta, but not the LHbeta, expression between groups or the different expression of ER´s and VTG between the FM20 and FM30 and these with respect to controls. It would be relevant to understand potential effects over puberty and initial gonad development.
  2. Should revise and be accurate with the reference to hormones; for example, the identity of the GnRH, as mentioned in a previous comment, or to cite inaccurately FSHbeta, LHbeta, ER´s or VTG as hormones, being subunits, receptors of proteins.
  3. English should be improved through the manuscript; for example, first line of the simple summary.

Author Response

Dear reviewer,

The authors appreciate your revision on giving us valuable commentary and suggestions in order to improve our work. Please find our responses to the file below.

Reviewer 2 Report

The Authors of the manuscript entitled "Effect of fishmeal content in the diet on the growth and sexual maturation of olive flounder (Paralichthys olivaceus) at a typical fish farm" have presented a paper describing a relatively standard study in the field of fisheries, focused on the effects of nutrition on gonadal development of a quite popular cultured fish species.

In my opinion, the Authors have committed a crucial mistake when composing their experimental diets and I believe that therefore this manuscript would be rejected from typical aquaculture-based journals, as it would not fit into the usual framework suggested for nutrition-oriented studies. However, given the amount of sheer work and effort put into this study, I believe that the Authors deserve the chance to publish their results, but only after correcting the mistakes I have outlined below, paragraph by paragraph. Indeed, there is a lot of stuff that needs to be improved.

Simple summary: Apart from the mistakes outlined below, what is missing here is a clear indication about the dietary replacement of fishmeal. Please specify what was it.
Line 20: Change "diet increasing" to "diets in aquaculture increases".
Line 22: Add "diets" after "low-fishmeal".
Line 23: What does "laboratory level" mean? Was this supposed to indicate an experimental hatchery or rearing facility? And how is "most flounder aquaculture" conducted in a laboratory environment? Please correct this statement.
Lines 23-25: This sentence, in its current structure, indicates that "The feed was farmed for 20 weeks", which of course makes no sense. The fish were farmed for 20 weeks, using the experimental feeds. Please correct this mistake.
Lines 26-27: Delete "it showed" and add "was observed" at the end of this sentence".
Line 29: Change "maturity" with "sexual development of olive flounder".

Abstract: Like in the simple summary, there is no information given about the dietary content which replaced the fishmeal.
Furthermore, according to the journal's guidelines "The abstract should be a total of about 200 words maximum", however, in its current version it is about 300 words. Regardless of this review, I believe the Editor would ask the Authors to shorten it, anyway.
Line 34: Correct to "-gonadal".
Line 40: Once again, the Authors insist on using the term "laboratory", which seems inadequate in this context. Please change that. Also, delete "From the results,".
Line 48: Change "diet" to "such high dietary protein".

Keywords: I suggest to change "diet replacement" with "fishmeal replacement" and make plural "hormones".

Introduction: Apart from the mistakes and problems listed below, I believe that this part was written decently.
Line 54: Change "support" to "sustain".
Lines 60-64: What I believe is missing here (between these 2 sentences) is a proper introduction of the topic of fishmeal in fish feeds (max. 2 new sentences). Right now, there is no link given between reduced catch fishery and fish feed production. Please expand this part, with proper references to back it up.
Line 68: Change "The fish gonad develops" to "Fish gonads develop".
Lines 86-87: It is unclear what this last sentence of the paragraph was supposed to mean, please rewrite it.
Line 90: Again, this "laboratory" is mentioned. Please correct it as suggested before.

Materials and Methods: Some key omissions were made in this part of the manuscript, some of which are severe, but I must admit that the molecular laboratory analyses (gene expression) were described thoroughly.
Paragraph 2.1: The information about the experimental setup is lacking, as several questions can be raised here: What was the water depth/volume of each tank? Does "water turnover, 20 times per day" indicate a 2000% water exchange per day? Was this a flow-through or recirculatory system? What was the mean salinity of the water? What was the approximate daily % feeding ratio? Were the fish starved prior to sampling? How were the WG, SGR, CF, FCR, PER, FI calculated (equations?), since they appear in Table 4 but are not even mentioned in the M&M section? All of these questions need to be answered, therefore, this paragraph has to be more precise and elaborate. Furthermore, in line 106, I believe it was supposed to be "euthanized" instead of "anaesthetized", since gonads were dissected.
Paragraph 2.2: Here lies probably the biggest flaw of the study - the dietary content varies between the control and experimental diets in 9 different components! It is really impossible to draw credible conclusions from such a setup, as there are definitely numerous differences between the diets in terms of dietary amino acid proportions or HUFA and PUFA % among the crude lipid content. It wasn't just the levels of two dietary ingredients that were exchanged, as it is usually the case in proper aquaculture studies focused on dietary composition, but 9! Keeping the crude protein and lipid content on the same level in all three diets does not exactly solve this issue. While I understand that the Authors' approach to this study was to focus solely on applicability and common practices in fisheries (considering the size of the tanks and total number of fish), I still consider this to be a big mistake in the whole planning process.
Furthermore, it is nowhere stated that each feeding group consisted of a duplicate of tanks. This raises more questions regarding the whole body content analyses. Were only the final 20 week samples analyzed this way? How were these samples prepared for these analyses? Were whole fish homogenized and analyzed separately, or the remains of fish which had their organs extracted for histology and gene expression? Were the samples pooled somehow, or was each analysis conducted for each fish separately, thus giving n = 20 for every analysis? Also, in how many replicates were these content analyses performed for the three diets? Since there is some statistical analysis indicated in Table 1, there must have been some replicates to allow such calculations to be made. Please rework and expand this paragraph profoundly.
Paragraph 2.3: The description of histological measurements needs to be clarified. How many oocytes were measured per specimen or group at each sampling point? How do the "short and long diameters" translate into the results shown in Figure 3, where only one diameter is shown? Please correct this part.
Paragraph 2.4.1: Were these tissues extracted from all sampled females during every sampling? I believe this information could be more precise, since different sex ratios were obtained in each group, at different sampling time points. What it means in terms of both histological and gene expression analyses is that this would indicate a different n value for each group and sampling. However, later on in the Results in Figures 4-6, the n = 6 and only initial and 20 week samples are shown. This is confusing, since the amount and origins of analyzed samples is not indicated clearly here, in the M&M section. Furthermore were there no gonads at the beginning of the experiment to constitute a group "0"?
Table 3: This table contains qPCR primers, thus I believe it should be removed from the main text and turned into a supplementary file.
Lines 172-173: A commonly made mistake occurred here - the Authors have already described (paragraph 2.4.2) how reverse transcription (RT) was performed! I believe RT-PCR was supposed to indicate Real-Time PCR, and therefore, to avoid confusion, it is widely suggested throughout the scientific world to stick to the term quantitative PCR (qPCR) for the final gene expression analysis. Please change the abbreviation "RT-PCR" to "qPCR" throughout the manuscript, including (Supplementary) Table 3.

Results: 
Table 4: There is no indication of the n value.
Table 5:
The title states 22 weeks of feeding, instead of 20. Please correct or explain this. Also, again no indication of the n value (which is a callback to my complaints about paragraph 2.2).
Figure 1: These graphs are confusing and are not consistent with the description given above. How is the GSI of CON group higher after 16 weeks, when the letters on the graph are indifferent at this sampling point between all three groups? It is clearly shown that "b" changes from "c" at 8 weeks in each group. Moreover, are these letters indicating only differences between groups at each timepoint, or between timepoints too? If so, then a Two-Way ANOVA is required, not One-Way. What about the differences in growth at each sampling, were they analyzed statistically too? Why is there no indication of such analysis? I believe it would be for the the best to rework this Figure from scratch into two graphs, one showing only the body mass changes of all three groups, the other showing only the GSIs.
Paragraph 3.2: I think that a small Table could be made here, containing the information about sex proportions, in order to remove them from the text and thus highlight them in a more profound way (just write the numbers of males and females sampled each time). Also, how did the Authors observe "phosphorus" in oocyte nuclei using just the H&E stain?
Furthermore, following my earlier complaints about Paragraph 2.3, it can be asked how did the Authors assess the differences in oocyte nuclei sizes, if only the whole cell diameter was measured? Or was it actually the size of the nuclei that was measured? Please clarify this issue.
Figure 2: There is no need to repeat the description of the same abbreviations (FC, OC) for each group, as they remain the same. Just indicate them once, at the end.
Figure 3: Please indicate the n of the measurements and indicate the applied statistical analysis (significance) - why is there a "*", when earlier the Authors used letters "abc" for the same ANOVA calculations? Again, was this One-way or Two-way ANOVA?
Figure 4: The Authors mention "different letters", but there is only one "*" sign and no indication of significance between the initial and CON and FM20 values for PSS-I expression, as it was mentioned above in the text. Please fix this issue. In comparison, Figures 5 and 6 are correct in this regard.

Discussion: Once again, in line 307, the Authors state they observed "phosphorus" in oocyte nuclei. How was this possible? Please add references that would prove that such observation can be made using simple H&E slides. Furthermore, the Authors did comment on the nuclei size of oocytes, even though it was not indicated in the M&Ms, that this parameter was actually measured. Once more, there are no citations to comment on such measurements.

Author Response

(The authors gave the same response as above.)

Round 2

Reviewer 1 Report

The authors have made an exhaustive revision of the original manuscript, adding the required lacking information and modifying adequately most, but not all, of the comments addressed, including additional changes. Nevertheless, there are still some details that need to be solved (see following comments) and also, the English style should still be improved. Thus, I consider correct this revision in general and the manuscript could be accepted after appropriate modifications.

Specific comments to major points raised in the previous report:

  1. NOT ADEQUATE YET. In the response, the authors say that they analysed the seabream GnRH and respond that “In regard to your advice, the supplement was made by adding more information to clear the idea of GnRH. In the scope of this study, sea-bream-type GnRH was investigated to elucidate the sexual maturation in olive flounder (Pham et al., 2006; Xu et al., 2012)”. Now we know which gene they analysed, but it is still incorrectly named in the manuscript. The authors must cite adequately this gene/hormone through the manuscript, as sbGnRH or GNRH-1, but not GnRH. Please, correct.
  2. OK
  3. NOT ADEQUATE YET. It is still not clearly addressed or missing, in section “material and methods”, the sampling timing and “n” for the qPCR analysis. As I mentioned before, this is clearly described for the previous parameters described in the section, but not for this ones and it still remains as it is in the revised version. The authors added that “only females were used in this analysis”, but still they do not mention the timing of the sampling (0-8-16-20 weeks), neither the n. Also, it is still confusing the “n” through the manuscript and I can mention two examples. First, in fig 2 they say in the legend that n=20 and thus the reader would assume that these data are from males and females together (this would have to be said in the legend!!!!), but I am wondering if GSI data (fig 2A) are really for a mix males and females, which would be weird. Second, in fig 4 they say in the legend “n=100”, which does not seem correct. They said somewhere that they measured 100 oocytes per individual, but this is not the “n” for the figure data. Please, correct. Another point within this comment, that I raised in the previous report, is why the authors did not make and/or include qPCR data for the 8 and 16 weeks. The authors responded that they actually performed these analyses but decided not to include them in the manuscript. It is weird to me and still think that the work would be more informative if this information would be included.
  4. OK
  5. NOT ADEQUATE YET. In the response the authors say that “In the abstract, the authors totally agree that ERalfa is not higher in FM20 than control and it is corrected by deleted from the abstract. However, The FSHbeta still is correct as it is significantly higher in FM20 than the control regarding to the figure 5. Thus, the authors decided to keep it in the abstract.”. I cannot understand why the authors say that, at 20 weeks, the FSHbeta is higher in FM20 than CON, if we can see in the figure that bars for CON and FM20 are superscripted with “ab” and “a”, indicating NO significant differences between them. Please, clarify your response and modify the manuscript accordingly.

Specific comments to responses to minor points raised in the previous report:

  1. OK
  2. OK
  3. NOT ADEQUATE YET. The English grammar and style should still be improved through the manuscript.

Author Response

The authors very appreciate that our work can be potentially considered for publication which is contingent upon our responses. Please find our responses to the comments below.

Reviewer 2 Report

The Authors of the manuscript entitled "Effect of fishmeal content in the diet on the growth and sexual maturation of olive flounder (Paralichthys olivaceus) at a typical fish farm" have listened to the numerous remarks of both reviewers and have put forth a significant effort to improve their paper accordingly. However, there still remain a few mistakes (some of which were newly introduced during the revision), mostly in the Materials and Methods section. As previously, I wish to respond to the applied changes in an organized manner, paragraph by paragraph.

Simple summary: The applied corrections significantly improved this paragraph and it is now clear, containing all the necessary information.

Abstract: Likewise above, the Authors have put a better emphasis on the explanation of the aims of the study. Nicely done.

Keywords: They appear to be fine now.

Introduction: The Authors have significantly improved this part, following closely the commentary of both reviewers. Minor corrections are only needed and I have outlined them below.
Line 55: Add "are reported" after "advantages".
Line 56:
I see there must be some linguistic problem going on here as the Authors repeat this mistake on other occasions in the manuscript as well. Namely, once again they have confused the word "nucleolus" (https://ko.wikipedia.org/wiki/%ED%95%B5%EC%86%8C%EC%B2%B4), which is a sub-organelle of every eukaryotic cell's nucleus (responsible for the production of rRNA and assembly of ribosome subunits), with "phosphorus" (https://ko.wikipedia.org/wiki/%EC%9D%B8), which is of course the mineral element (15P) that the Authors obviously wanted to mention here. I hope my explanation will save them from committing the same mistake in future papers.

Materials and Methods: The description of the experimental setup and of the conducted analyses was much improved. Especially, I very much enjoyed the addition of the equations and Figure 1.
However, I have found two issues which are still lingering:
Paragraph 2.1: Although the Authors remarked in their response to the reviewers that the information about the "n" number of samples for histological and qPCR analyses was clarified, this issue was not fully solved. I suggest to add a sentence for clarification, in the current line 112, in front of the one starting with "The equations ...". The sentence should be something like this: "Tissue samples were dissected from these fish and fixed for histological (gonads) and qPCR (brain, pituitary, liver) analyses, accordingly".
Paragraph 2.3: In their response, the Authors mentioned that 100 oocytes were measured in each female specimen, what would result in different "n" values for every group, as each time there were different amounts of sampled females in every group. However, later on in Figure 4 there is "n=100". The information about the morphometrics needs to be better outlined here in M&Ms. Either it was 100 oocytes/specimen and different "n" values each time, or it was 100 oocytes per group at each sampling time point, and the number of measured cells per specimen was adjusted for the number of females. Or maybe 10 females were measured each time, like in qPCR analyses, resulting in a total n=1000? Please clarify and add the necessary information.

Beside the aforementioned issues, I only found some typos in the newly added text which need to be corrected.
Line 98: I believe it should rather be "10 x 10 m (100 m2)" or simply "10 x 10 meters".
Line 104: Correct typo to "exchanged".
Line 105: Correct typos to "maintained" and "salinity".
Line 112: Make plural to "equations".
Line 126: Make plural to "changes".
Line 127: Correct typos to "composition" and "flounders". Add "additional" before "20 fish" and "after 20 weeks" after "selected".
Line 128: Correct typo to "analysis". Once again, "euthanized" instead of "anesthetized".
Line 129: Add a comma after "Following".
Line 130: Correct typo to "composition".
Line 162: Add "(10 per group)" at the end of the sentence.

Results: The applied changes to the graphs in Figure 2 have fixed the previous problems. Other than the typos below, this part appears to be adequate now.
Line 246: Add "at each sampling time point" after "groups".
Lines 253-254:
Correct the plural forms to "nucleoli" and "nuclei".

Discussion: Other than the returning linguistic problems shown below, I do not have any major issues with this paragraph.
Line 335: Correct the plural form to "nucleoli".
Lines 338-339: Again, there is a problem with these "nucleoli" and "nuclei" which impedes the understanding of this sentence. Correct me if I'm wrong, but I would say that during fish gonadal maturation, the volumes of both the nucleus and the cytoplasm of oocytes grow concurrently. Therefore, what "loss of a large amount" was actually observed? Was it the reduction fo the numbers of nucleoli in the oocyte nuclei? Furthermore, the"migration" of what exact structures was observed? Of nucleoli inside the nuclei, or of the nuclei inside the cytoplasm, along with the accumulationg of egg yolk? Please fix this sentence and use proper plural forms of "nucleoli" and "nuclei".
Line 370: Change "containing" to "with attached" and "nucleous" to "phosphates" (even the abstract of your provided reference no. 47 clearly states such information about Vtg being a "phospholipoglycoprotein").

Conclusions: Were already correct in the previous draft.

Author Response

(The authors gave the same response as above.)
